# Alternating Multi-bit Quantization for Recurrent Neural Networks

**Chen Xu**[1,*] **Jianqiang Yao**[2], **Zhouchen Lin**[1,3,†] **Wenwu Ou**[2], **Yuanbin Cao**[4], **Zhirong Wang**[2], **Hongbin Zha**[1,3]

[1] Key Laboratory of Machine Perception (MOE), School of EECS, Peking University, China
[2] Search Algorithm Team, Alibaba Group, China
[3] Cooperative Medianet Innovation Center, Shanghai Jiao Tong University, China
[4] AI-LAB, Alibaba Group, China

`xuen@pku.edu.cn,tianduo@taobao.com,zlin@pku.edu.cn,santong.oww@taobao.com`
`lingzun.cyb@alibaba-inc.com, qingfeng@taobao.com,zha@cis.pku.edu.cn`

## Abstract

Recurrent neural networks have achieved excellent performance in many applications. However, on portable devices with limited resources, the models are often too large to deploy. For applications on the server with large scale concurrent requests, the latency during inference can also be very critical for costly computing resources. In this work, we address these problems by quantizing the network, both weights and activations, into multiple binary codes $\{-1, +1\}$. We formulate the quantization as an optimization problem. Under the key observation that once the quantization coefficients are fixed the binary codes can be derived efficiently by binary search tree, alternating minimization is then applied. We test the quantization for two well-known RNNs, i.e., long short term memory (LSTM) and gated recurrent unit (GRU), on the language models. Compared with the full-precision counter part, by 2-bit quantization we can achieve $\sim 16\times$ memory saving and $\sim 6\times$ real inference acceleration on CPUs, with only a reasonable loss in the accuracy. By 3-bit quantization, we can achieve almost no loss in the accuracy or even surpass the original model, with $\sim 10.5\times$ memory saving and $\sim 3\times$ real inference acceleration. Both results beat the exiting quantization works with large margins. We extend our alternating quantization to image classification tasks. In both RNNs and feedforward neural networks, the method also achieves excellent performance.

## 1 Introduction

Recurrent neural networks (RNNs) are specific type of neural networks which are designed to model the sequence data. In last decades, various RNN architectures have been proposed, such as Long-Short-Term Memory (LSTM) (Hochreiter & Schmidhuber, 1997) and Gated Recurrent Units Cho et al. (2014). They have enabled the RNNs to achieve state-of-art performance in many applications, e.g., language models (Mikolov et al., 2010), neural machine translation (Sutskever et al., 2014; Wu et al., 2016), automatic speech recognition (Graves et al., 2013), image captions (Vinyals et al., 2015), etc. However, the models often build on high dimensional input/output,e.g., large vocabulary in language models, or very deep inner recurrent networks, making the models have too many parameters to deploy on portable devices with limited resources. In addition, RNNs can only be executed sequentially with dependence on current hidden states. This causes large latency during inference. For applications in the server with large scale concurrent requests, e.g., on-line machine translation and speech recognition, large latency leads to limited requests processed per machine to meet the stringent response time requirements. Thus much more costly computing resources are in demand for RNN based models.

To alleviate the above problems, several techniques can be employed, i.e., low rank approximation (Sainath et al., 2013; Jaderberg et al., 2014; Lebedev et al., 2014; Tai et al., 2016), sparsity (Liu

---

*Work performed while interning at Alibaba search algorithm team.
†Corresponding author.

et al., 2015; Han et al., 2015; 2016; Wen et al., 2016), and quantization. All of them are build on the redundancy of current networks and can be combined. In this work, we mainly focus on quantization based methods. More precisely, we are to quantize all parameters into multiple binary codes $\{-1, +1\}$.

The idea of quantizing both weights and activations is firstly proposed by (Hubara et al., 2016a). It has shown that even 1-bit binarization can achieve reasonably good performance in some visual classification tasks. Compared with the full precision counterpart, binary weights reduce the memory by a factor of 32. And the costly arithmetic operations between weights and activations can then be replaced by cheap XNOR and bitcount operations(Hubara et al., 2016a), which potentially leads to much acceleration. Rastegari et al. (2016) further incorporate a real coefficient to compensate for the binarization error. They apply the method to the challenging ImageNet dataset and achieve better performance than pure binarization in (Hubara et al., 2016a). However, it is still of large gap compared with the full precision networks. To bridge this gap, some recent works (Hubara et al., 2016b; Zhou et al., 2016; 2017) further employ quantization with more bits and achieve plausible performance. Meanwhile, quite an amount of works, e.g., (Courbariaux et al., 2015; Li et al., 2016; Zhu et al., 2017; Guo et al., 2017), quantize the weights only. Although much memory saving can be achieved, the acceleration is very limited in modern computing devices (Rastegari et al., 2016).

Among all existing quantization works, most of them focus on convolutional neural networks (CNNs) while pay less attention to RNNs. As mentioned earlier, the latter is also very demanding. Recently, (Hou et al., 2017) showed that binarized LSTM with preconditioned coefficients can achieve promising performance in some easy tasks such as predicting the next character. However, for RNNs with large input/output, e.g., large vocabulary in language models, it is still very challenging for quantization. Both works of Hubara et al. (2016b) and Zhou et al. (2017) test the effectiveness of their multi-bit quantized RNNs to predict the next word. Although using up to 4-bits, the results with quantization still have noticeable gap with those with full precision. This motivates us to find a better method to quantize RNNs. The main contribution of this work is as follows:

(a) We formulate the multi-bit quantization as an optimization problem. The binary codes $\{-1, +1\}$ are learned instead of rule-based. For the first time, we observe that the codes can be optimally derived by the binary search tree once the coefficients are knowns in advance, see, e.g., Algorithm 1. Thus the whole optimization is eased by removing the discrete unknowns, which are very difficult to handle.

(b) We propose to use alternating minimization to tackle the quantization problem. By separating the binary codes and real coefficients into two parts, we can solve the subproblem efficiently when one part is fixed. With proper initialization, we only need two alternating cycles to get high precision approximation, which is effective enough to even quantize the activations on-line.

(c) We systematically evaluate the effectiveness of our alternating quantization on language models. Two well-known RNN structures, i.e., LSTM and GRU, are tested with different quantization bits. Compared with the full-precision counterpart, by 2-bit quantization we can achieve $\sim 16 \times$ memory saving and $\sim 6 \times$ real inference acceleration on CPUs, with a reasonable loss on the accuracy. By 3-bit quantization, we can achieve almost no loss in accuracy or even surpass the original model with $\sim 10.5 \times$ memory saving and $\sim 3 \times$ real inference acceleration. Both results beat the exiting quantization works with large margins. To illustrate that our alternating quantization is very general to extend, we apply it to image classification tasks. In both RNNs and feedforward neural networks, the technique still achieves very plausible performance.

## 2 EXISTING MULTI-BIT QUANTIZATION METHODS

Before introducing our proposed multi-bit quantization, we first summarize existing works as follows:

(a) Uniform quantization method (Rastegari et al., 2016; Hubara et al., 2016b) firstly scales its value in the range $x \in [-1, 1]$. Then it adopts the following $k$-bit quantization:

$$q_k(x) = 2 \left( \frac{\text{round}[(2^k - 1)(\frac{x+1}{2})]}{2^k - 1} - \frac{1}{2} \right), \tag{1}$$

after which the method scales back to the original range. Such quantization is rule based thus is very easy to implement. The intrinsic benefit is that when computing inner product

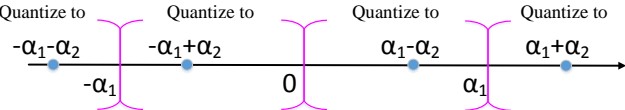

Figure 1: Illustration of the optimal 2-bit quantization when $\alpha_1$ and $\alpha_2$ ($\alpha_1 \geq \alpha_2$) are known in advance. The values are quantized into $-\alpha_1 - \alpha_2$, $-\alpha_1 + \alpha_2$, $\alpha_1 - \alpha_2$, and $\alpha_1 + \alpha_2$, respectively. And the partition intervals are optimally separated by the middle points of adjacent quantization codes, i.e., $-\alpha_1$, $0$, and $\alpha_1$, correspondingly.

of two quantized vectors, it can employ cheap bit shift and count operations to replace costly multiplications and additions operations. However, the method can be far from optimum when quantizing non-uniform data, which is believed to be the trained weights and activations of deep neural network (Zhou et al., 2017).

(b) Balanced quantization (Zhou et al., 2017) alleviates the drawbacks of the uniform quantization by firstly equalizing the data. The method constructs $2^k$ intervals which contain roughly the same percentage of the data. Then it linearly maps the center of each interval to the corresponding quantization code in (1). Although sounding more reasonable than the uniform one, the affine transform on the centers can still be suboptimal. In addition, there is no guarantee that the evenly spaced partition is more suitable if compared with the non-evenly spaced partition for a specific data distribution.

(c) Greedy approximation (Guo et al., 2017) instead tries to learn the quantization by tackling the following problem:

$$\min_{\{\alpha_i, \mathbf{b}_i\}_{i=1}^k} \left\| \mathbf{w} - \sum_{i=1}^k \alpha_i \mathbf{b}_i \right\|^2, \quad \text{with} \quad \mathbf{b}_i \in \{-1, +1\}^n. \tag{2}$$

For $k = 1$, the above problem has a closed-form solution (Rastegari et al., 2016). Greedy approximation extends to $k$-bit ($k > 1$) quantization by sequentially minimizing the residue. That is

$$\min_{\alpha_i, \mathbf{b}_i} \left\| \mathbf{r}_{i-1} - \alpha_i \mathbf{b}_i \right\|^2, \quad \text{with} \quad \mathbf{r}_{i-1} = \mathbf{w} - \sum_{j=1}^{i-1} \alpha_j \mathbf{b}_j. \tag{3}$$

Then the optimal solution is given as

$$\alpha_i = \frac{1}{n} \|\mathbf{r}_{i-1}\|_1 \quad \text{and} \quad \mathbf{b}_i = \text{sign}(\mathbf{r}_{i-1}). \tag{4}$$

Greedy approximation is very efficient to implement in modern computing devices. Although not able to reach a high precision solution, the formulation of minimizing quantization error is very promising.

(d) Refined greedy approximation (Guo et al., 2017) extends to further decrease the quantization error. In the $j$-th iteration after minimizing problem (3), the method adds one extra step to refine all computed $\{\alpha_i\}_{i=1}^j$ with the least squares solution:

$$[\alpha_1, \ldots, \alpha_j] = \left( (\mathbf{B}_j^T \mathbf{B}_j)^{-1} \mathbf{B}_j^T \mathbf{w} \right)^T, \quad \text{with} \quad \mathbf{B}_j = [\mathbf{b}_1, \ldots, \mathbf{b}_j], \tag{5}$$

In experiments of quantizing the weights of CNN, the refined approximation is verified to be better than the original greedy one. However, as we will show later, the refined method is still far from satisfactory for quantization accuracy.

Besides the general multi-bit quantization as summarized above, Li et al. (2016) propose ternary quantization by extending 1-bit binarization with one more feasible state, 0. It does quantization by tackling $\min_{\alpha, \mathbf{t}} \|\mathbf{w} - \alpha \mathbf{t}\|_2^2$ with $\mathbf{t} \in \{-1, 0, +1\}^n$. However, no efficient algorithm is proposed in (Li et al., 2016). They instead empirically set the entries $w$ with absolute scales less than $0.7/n\|\mathbf{w}\|_1$ to 0 and binarize the left entries as (4). In fact, ternary quantization is a special case of the 2-bit quantization in (2), with an additional constraint that $\alpha_1 = \alpha_2$. When the binary codes are fixed, the optimal coefficient $\alpha_1$ (or $\alpha_2$) can be derived by least squares solution similar to (5).

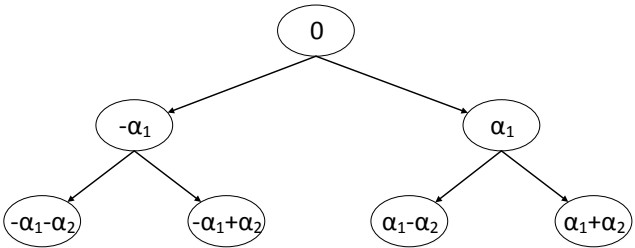

Figure 2: Illustration of binary search tree to determine the optimal quantization.

---

**Algorithm 1:** Binary Search Tree (BST) to determine to optimal code

---

**BST**$(w, \mathbf{v})$
{$w$ is the real value to be quantized}
{$\mathbf{v}$ is the vector of quantization codes in ascending order}

1    $m = \text{length}(\mathbf{v})$
2    **if** $m == 1$ **then**
3    |    **return** $\mathbf{v}_1$
4    **end**
5    **if** $w \geq (v_{m/2} + v_{m/2+1})/2$ **then**
6    |    **BST**$(w, \mathbf{v}_{m/2+1:m})$
7    **else**
8    |    **BST**$(w, \mathbf{v}_{1:m/2})$
9    **end**

---

In parallel to the binarized quantization discussed here, vector quantization is applied to compress the weights for feedforward neural networks (Gong et al., 2014; Han et al., 2016). Different from ours where all weights are directly constraint to $\{-1, +1\}$, vector quantization learns a small codebook by applying k-means clustering to the weights or conducting product quantization. The weights are then reconstructed by indexing the codebook. It has been shown that by such a technique, the number of parameters can be reduced by an order of magnitude with limited accuracy loss (Gong et al., 2014). It is possible that the multi-bit quantized binary weight can be further compressed by using the product quantization.

## 3    OUR ALTERNATING MULTI-BIT QUANTIZATION

Now we introduce our quantization method. We tackle the same minimization problem as (2). For simplicity, we firstly consider the problem with $k = 2$. Suppose that $\alpha_1$ and $\alpha_2$ are known in advance with $\alpha_1 \geq \alpha_2 \geq 0$, then the quantization codes are restricted to $\mathbf{v} = \{-\alpha_1 - \alpha_2, -\alpha_1 + \alpha_2, \alpha_1 - \alpha_2, \alpha_1 + \alpha_2\}$. For any entry $w$ of $\mathbf{w}$ in problem (2), its quantization code is determined by the least distance to all codes. Consequently, we can partition the number axis into $4$ intervals. And each interval corresponds to one particular quantization code. The common point of two adjacent intervals then becomes the middle point of the two quantization codes, i.e., $-\alpha_1$, $0$, and $\alpha_1$. Fig. 1 gives an illustration.

For the general $k$-bit quantization, suppose that $\{\alpha_i\}_{i=1}^k$ are known and we have all possible codes in ascending order, i.e., $\mathbf{v} = \{-\sum_{i=1}^k \alpha_i, \ldots, \sum_{i=1}^k \alpha_i\}$. Similarly, we can partition the number axis into $2^k$ intervals, in which the boundaries are determined by the centers of two adjacent codes in $\mathbf{v}$, i.e., $\{(v_i + v_{i+1})/2\}_{i=1}^{2^k-1}$. However, directly comparing per entry with all the boundaries needs $2^k$ comparisons, which is very inefficient. Instead, we can make use of the ascending property in $\mathbf{v}$. Hierarchically, we partition the codes of $\mathbf{v}$ evenly into two ordered sub-sets, i.e., $\mathbf{v}_{1:m/2}$ and $\mathbf{v}_{m/2+1:m}$ with $m$ defined as the length of $\mathbf{v}$. If $w < (v_{m/2} + v_{m/2+1})/2$, its feasible codes are then optimally restricted to $\mathbf{v}_{1:m/2}$. And if $w \geq (v_{m/2} + v_{m/2+1})/2$, its feasible codes become $\mathbf{v}_{m/2+1:m}$. By recursively evenly partition the ordered feasible codes, we can then efficiently determine the

---

**Algorithm 2:** Alternating Multi-bit Quantization

---

**Require :** Full precision weight $\mathbf{w} \in \mathbb{R}^n$, number of bits $k$, total iterations $T$
**Ensure  :** $\{\alpha_i, \mathbf{b}_i\}_{i=1}^k$

1  Greedy Initialize $\{\alpha_i, \mathbf{b}_i\}_{i=1}^k$ as (4)
2  **for** iter $\leftarrow 1$ **to** $T$ **do**
3  $\quad$ Update $\{\alpha_i\}_{i=1}^k$ as (5)
4  $\quad$ Construct $\mathbf{v}$ of all feasible codes in accending order
5  $\quad$ Update $\{\mathbf{b}_i\}_{i=1}^k$ as Algorithm 1.
6  **end**

---

optimal code for per entry by only $k$ comparisons. The whole procedure is in fact a binary search tree. We summarize it in Algorithm 1. Note that once getting the quantization code, it is straightforward to map to the binary code $\mathbf{b}$. Also, by maintaining a mask vector with the same size as $\mathbf{w}$ to indicate the partitions, we could operate BST for all entries simultaneously. To give a better illustration, we give a binary tree example for $k = 2$ in Fig. 2. Note that for $k = 2$, we can even derive the optimal codes by a closed form solution, i.e., $\mathbf{b}_1 = \text{sign}(\mathbf{w})$ and $\mathbf{b}_2 = \text{sign}(\mathbf{w} - \alpha_1 \mathbf{b}_1)$ with $\alpha_1 \geq \alpha_2 \geq 0$.

Under the above observation, let us reconsider the refined greedy approximation (Guo et al., 2017) introduced in Section 2. After modification on the computed $\{\alpha_i\}_{i=1}^j$ as (5), $\{\mathbf{b}_i\}_{i=2}^j$ are no longer optimal while the method keeps all of them fixed. To improve the refined greedy approximation, alternating minimizing $\{\alpha_i\}_{i=1}^k$ and $\{\mathbf{b}_i\}_{i=1}^k$ becomes a natural choice. Once getting $\{\mathbf{b}_i\}_{i=1}^k$ as described above, we can optimize $\{\alpha_i\}_{i=1}^k$ as (5). In real experiments, we find that by greedy initialization as (4), only two alternating cycles is good enough to find high precision quantization. For better illustration, we summarize our alternating minimization in Algorithm 2. For updating $\{\alpha_i\}_{i=1}^k$, we need $2k^2n$ binary operations and $kn$ non-binary operations. Combining $kn$ non-binary operations to determine the binary code, for total $T$ alternating cycles, we thus need $2Tk^2n$ binary operations and $2(T + 1)kn$ non-binary operations to quantize $\mathbf{w} \in \mathbb{R}^n$ into $k$-bit, with the extra $2kn$ corresponding to greedy initialization.

## 4 APPLY ALTERNATING MULTI-BIT QUANTIZATION TO RNNS

**Implementation**. We firstly introduce the implementation details for quantizing RNN. For simplicity, we consider the one layer LSTM for language model. The goal is to predict the next word indexed by $t$ in a sequence of one-hot word tokens $(y_1^*, \ldots, y_N^*)$ as follows:

$$
\begin{aligned}
\mathbf{x}_t &= \mathbf{W}_e^T \mathbf{y}_{t-1}^*, \\
\mathbf{i}_t, \mathbf{f}_t, \mathbf{o}_t, \mathbf{g}_t &= \sigma(\mathbf{W}_i \mathbf{x}_t + \mathbf{b}_i + \mathbf{W}_h \mathbf{h}_{t-1} + \mathbf{b}_h), \\
\mathbf{c}_t &= \mathbf{f}_t \odot \mathbf{c}_{t-1} + \mathbf{i}_t \odot \mathbf{g}_t, \quad \mathbf{h}_t = \mathbf{o}_t \odot \tanh(\mathbf{c}_t), \\
\mathbf{y}_t &= \text{softmax}(\mathbf{W}_s \mathbf{h}_t + \mathbf{b}_s).
\end{aligned}
\tag{6}
$$

where $\sigma$ represents the activation function. In the above formulation, the multiplication between the weight matrices and the vectors $\mathbf{x}_t$ and $\mathbf{h}_t$ occupy most of the computation. This is also where we apply quantization to. For the weight matrices, We do not apply quantization on the full but rather row by row. During the matrix vector product, we can firstly execute the binary multiplication. Then element-wisely multiply the obtained binary vector with the high precision scaling coefficients. Thus little extra computation results while much more freedom is brought to better approximate the weights. We give an illustration on the left part of Fig. 3. Due to one-hot word tokens, $\mathbf{x}_t$ corresponds to one specific row in the quantized $\mathbf{W}_e$. It needs no more quantization. Different from the weight matrices, $\mathbf{h}_t$ depends on the input, which needs to be quantized on-line during inference. For consistent notation with existing work, e.g., (Hubara et al., 2016b; Zhou et al., 2017), we also call quantizing on $\mathbf{h}_t$ as quantizing on activation.

For $\mathbf{W} \in \mathbb{R}^{m \times n}$ and $\mathbf{h}_t \in \mathbb{R}^n$, the standard matrix-vector product needs $2mn$ operations. For the quantized product between $k_w$-bit $\mathbf{W}$ and $k_h$-bit $\mathbf{h_t}$, we have $2k_w k_h mn + 4k_h^2 n$ binary operations and $6k_h n + 2k_w k_h m$ non-binary operations, where $6k_h n$ corresponds to the cost of alternating approximation ($T = 2$) and $2k_w k_h m$ corresponds to the final product with coefficients. As the binary

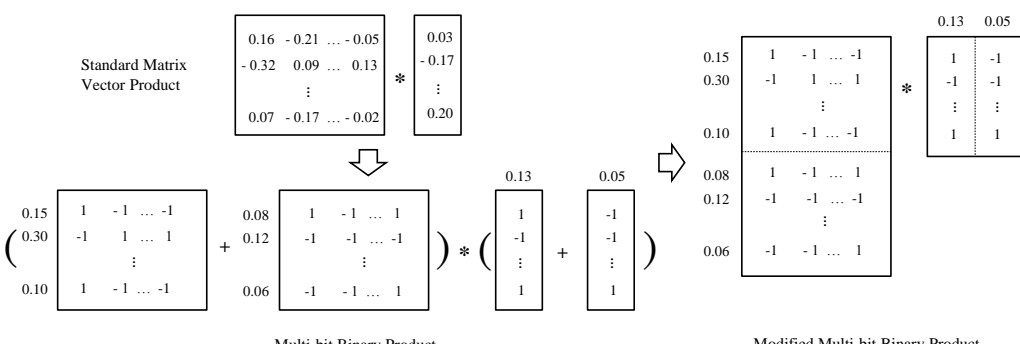

Figure 3: Illustration of quantized matrix vector multiplication (left part). The matrix is quantized row by row, which provides more freedom to approximate while adds little extra computation. By reformulating as the right part, we can make full use of the intrinsic parallel binary matrix vector multiplication for further acceleration.

multiplication operates in 1 bit, whereas the full precision multiplication operates in 32 bits, despite the feasible implementations, the acceleration can be $32\times$ in theory. For alternating quantization here, the overall theoretical acceleration is thus computed as $\gamma = \frac{2mn}{\frac{1}{32}(2k_w k_h mn + 4k_n^2 n) + 6k_h n + 2k_w k_h m}$. Suppose that LSTM has hidden states $n = 1024$, then we have $\mathbf{W}_h \in \mathbb{R}^{4096 \times 1024}$. The acceleration ratio becomes roughly $7.5\times$ for $(k_h, k_w) = (2, 2)$ and $3.5\times$ for $(k_h, k_w) = (3, 3)$. In addition to binary operations, the acceleration in real implementations can be largely affected by the size of the matrix, where much memory reduce can result in better utilizing in the limited faster cache. We implement the binary multiplication kernel in CPUs. Compared with the much optimized Intel Math Kernel Library (MKL) on full precision matrix vector multiplication, we can roughly achieve $6\times$ for $(k_h, k_w) = (2, 2)$ and $3\times$ for $(k_h, k_w) = (3, 3)$. For more details, please refer to Appendix A.

As indicated in the left part of Fig. 3, the binary multiplication can be conducted sequentially by associativity. Although the operation is suitable for parallel computing by synchronously conducting the multiplication, this needs extra effort for parallelization. We instead concatenate the binary codes as shown in the right part of Fig. 3. Under such modification, we are able to make full use of the much optimized inner parallel matrix multiplication, which gives the possibility for further acceleration. The final result is then obtained by adding all partitioned vectors together, which has little extra computation.

**Training.** As firstly proposed by Courbariaux et al. (2015), during the training of quantized neural network, directly adding the moderately small gradients to quantized weights will result in no change on it. So they maintain a full precision weight to accumulate the gradients then apply quantization in every mini-batch. In fact, the whole procedure can be mathematically formulated as a bi-level optimization (Colson et al., 2007) problem:

$$\min_{\mathbf{w}, \{\alpha_i, \mathbf{b}_i\}_{i=1}^k} f\left(\sum_{i=1}^k \alpha_i \mathbf{b}_i\right)$$
$$s.t. \quad \{\alpha_i, \mathbf{b}_i\}_{i=1}^k = \arg\min_{\{\alpha_i', \mathbf{b}_i'\}_{i=1}^k} \left\|\mathbf{w} - \sum_{i=1}^k \alpha_i' \mathbf{b}_i'\right\|^2. \tag{7}$$

Denote the quantized weight as $\hat{\mathbf{w}} = \sum_{i=1}^k \alpha_i \mathbf{b}_i$. In the forward propagation, we derive $\hat{\mathbf{w}}$ from the full precision $\mathbf{w}$ in the lower-level problem and apply it to the upper-level function $f(\cdot)$, i.e., RNN in this paper. During the backward propagation, the derivative $\frac{\partial f}{\partial \hat{\mathbf{w}}}$ is propagated back to $\mathbf{w}$ through the lower-level function. Due to the discreteness of $\mathbf{b}_i$, it is very hard to model the implicit dependence of $\hat{\mathbf{w}}$ on $\mathbf{w}$. So we also adopt the "straight-through estimate" as (Courbariaux et al., 2015), i.e., $\frac{\partial f}{\partial \mathbf{w}} = \frac{\partial f}{\partial \hat{\mathbf{w}}}$. To compute the derivative on the quantized hidden state $\mathbf{h}_t$, the same trick is applied. During the training, we find the same phenomenon as Hubara et al. (2016b) that some

Table 1: Measurement on the approximation of different quantization methods, e.g., Uniform (Hubara et al., 2016b), Balanced (Zhou et al., 2017), Greedy (Guo et al., 2017), Refined (Guo et al., 2017), and our Alternating method, see Section 2. We apply these methods to quantize the full precision pre-trained weight of LSTM on the PTB dataset. The best values are in bold. W-bits represents the number of weight bits and FP denotes full precision.

| Relative MSE | | | | Testing PPW | | | |
|---|---|---|---|---|---|---|---|
| W-Bits | 2 | 3 | 4 | 2 | 3 | 4 | FP |
| Uniform | 1.070 | 0.404 | 0.302 | 283.2 | 227.3 | 216.3 | |
| Balanced | 0.891 | 0.745 | 0.702 | 10287.6 | 9106.4 | 8539.8 | |
| Greedy | 0.146 | 0.071 | 0.042 | 118.9 | 99.4 | 95.0 | 89.8 |
| Refined | 0.137 | 0.060 | 0.030 | 105.3 | 95.4 | 93.1 | |
| Alternating (ours) | **0.125** | **0.043** | **0.019** | **103.1** | **93.8** | **91.4** | |

Table 2: Quantization on the full precision pre-trained weight of GRU on the PTB dataset.

| Relative MSE | | | | Testing PPW | | | |
|---|---|---|---|---|---|---|---|
| W-Bits | 2 | 3 | 4 | 2 | 3 | 4 | FP |
| Uniform | 6.138 | 3.920 | 3.553 | 3161906.6 | 771259.6 | 715781.9 | |
| Balanced | 1.206 | 1.054 | 1.006 | 2980.4 | 3396.3 | 3434.1 | |
| Greedy | 0.377 | 0.325 | 0.304 | 135.7 | 105.5 | 99.2 | 92.5 |
| Refined | 0.128 | 0.055 | 0.030 | 111.6 | 99.1 | 97.0 | |
| Alternating (ours) | **0.120** | **0.044** | **0.021** | **110.3** | **97.3** | **95.2** | |

entries of $\mathbf{w}$ can grow very large, which become outliers and harm the quantization. Here we simply clip $\mathbf{w}$ in the range of $[-1, 1]$.

## 5   EXPERIMENTS ON THE LANGUAGE MODELS

In this section, we conduct quantization experiments on language models. The two most well-known recurrent neural networks, i.e., LSTM (Hochreiter & Schmidhuber, 1997) and GRU (Cho et al., 2014), are evaluated. As they are to predict the next word, the performance is measured by perplexity per word (PPW) metric. For all experiments, we initialize with the pre-trained model and using vanilla SGD. The initial learning rate is set to 20. Every epoch we evaluate on the validation dataset and record the best value. When the validation error exceeds the best record, we decrease learning rate by a factor of 1.2. Training is terminated once the learning rate less than 0.001 or reaching the maximum epochs, i.e., 80. The gradient norm is clipped in the range $[-0.25, 0.25]$. We unroll the network for 30 time steps and regularize it with the standard dropout (probability of dropping out units equals to 0.5) (Zaremba et al., 2014). For simplicity of notation, we denote the methods using uniform, balanced, greedy, refined greedy, and our alternating quantization as Uniform, Balanced, Greedy, Refined, and Alternating, respectively.

**Peen Tree Bank.** We first conduct experiments on the Peen Tree Bank (PTB) corpus (Marcus et al., 1993), using the standard preprocessed splits with a 10K size vocabulary (Mikolov, 2012). The PTB dataset contains 929K training tokens, 73K validation tokens, and 82K test tokens. For fair comparison with existing works, we also use LSTM and GRU with 1 hidden layer of size 300. To have a glance at the approximation ability of different quantization methods as detailed in Section 2, we firstly conduct experiments by directly quantizing the trained full precision weight (neither quantization on activation nor retraining). Results on LSTM and GRU are shown in Table 1 and Table 2, respectively. The left parts record the relative mean squared error of quantized weight matrices with full precision one. We can see that our proposed Alternating can get much lower error across all

Table 3: Testing PPW of multi-bit quantized LSTM and GRU on the PTB dataset. W-Bits and A-Bits represent the number of weight and activation bits, respectively.

| | LSTM | | | | | GRU | | | | |
|---|---|---|---|---|---|---|---|---|---|---|
| W-Bits / A-Bits | 2/2 | 2/3 | 3/3 | 4/4 | FP/FP | 2/2 | 2/3 | 3/3 | 4/4 | FP/FP |
| Uniform | − | 220 | − | 100 | 97 | − | − | − | − | − |
| Balanced | 126 | 123 | − | 114 | 107 | 142 | − | − | 116 | 100 |
| Refined | 100.3 | 95.6 | 91.3 | − | 89.8 | 105.1 | 100.3 | 95.9 | − | 92.5 |
| Alternating (ours) | **95.8** | **91.9** | **87.9** | − | | **101.2** | **97.0** | **92.9** | − | |

Table 4: Testing PPW of multi-bit quantized LSTM and GRU on the WikiText-2 dataset.

| | LSTM | | | | GRU | | | |
|---|---|---|---|---|---|---|---|---|
| W-Bits / A-Bits | 2/2 | 2/3 | 3/3 | FP/FP | 2/2 | 2/3 | 3/3 | FP/FP |
| Refined | 108.7 | 105.8 | 102.2 | 100.1 | 117.2 | 114.1 | 111.8 | 106.7 |
| Alternating (ours) | **106.1** | **102.7** | **98.7** | | **113.7** | **110.2** | **106.4** | |

Table 5: Testing PPW of multi-bit quantized LSTM and GRU on the Text8 dataset.

| | LSTM | | | | GRU | | | |
|---|---|---|---|---|---|---|---|---|
| W-Bits / A-Bits | 2/2 | 2/3 | 3/3 | FP/FP | 2/2 | 2/3 | 3/3 | FP/FP |
| Refined | 135.6 | 122.3 | 110.2 | 101.1 | 135.8 | 126.9 | 118.3 | 111.6 |
| Alternating (ours) | **108.8** | **105.1** | **98.8** | | **124.5** | **118.7** | **114.0** | |

varying bit. We also measure the testing PPW for the quantized weight as shown in the right parts of Table 1 and 2. The results are in consistent with the left part, where less errors result in lower testing PPW. Note that Uniform and Balanced quantization are rule-based and not aim at minimizing the error. Thus they can have much worse result by direct approximation. We also repeat the experiment on other datasets. For both LSTM and GRU, the results are very similar to here.

We then conduct experiments by quantizing both weights and activations. We train with the batch size 20. The final result is shown in Table 3. Besides comparing with the existing works, we also conduct experiment for Refined as a competitive baseline. We do not include Greedy as it is already shown to be much inferior to the refined one, see, e.g., Table 1 and 2. As Table 3 shows, our full precision model can attain lower PPW than the existing works. However, when considering the gap between quantized model with the full precision one, our alternating quantized neural network is still far better than existing works, i.e., Uniform (Hubara et al., 2016b) and Balanced (Zhou et al., 2017). Compared with Refined, our Alternating quantization can achieve compatible performance using 1-bit less quantization on weights or activations. In other words, under the same tolerance of accuracy drop, Alternating executes faster and uses less memory than Refined. We can see that our 3/3 weights/activations quantized LSTM can achieve even better performance than full precision one. A possible explanation is due to the regularization introduced by quantization (Hubara et al., 2016b).

**WikiText-2** (Merity et al., 2017) is a dataset released recently as an alternative to PTB. It contains 2088K training, 217K validation, and 245K test tokens, and has a vocabulary of 33K words, which is roughly 2 times larger in dataset size, and 3 times larger in vocabulary than PTB. We train with one layer's hidden state of size 512 and set the batch size to 100. The result is shown in Table 4. Similar to PTB, our Alternating can use 1-bit less quantization to attain compatible or even lower PPW than Refined.

**Text8.** In order to determine whether Alternating remains effective with a larger dataset, we perform experiments on the Text8 corpus (Mikolov et al., 2014). Here we follow the same setting as (Xie et al., 2017). The first 90M characters are used for training, the next 5M for validation, and the final 5M for testing, resulting in 15.3M training tokens, 848K validation tokens, and 855K test tokens. We also preprocess the data by mapping all words which appear 10 or fewer times to the unknown token, resulting in a 42K size vocabulary. We train LSTM and GRU with one hidden layer of size 1024 and set the batch size to 100. The result is shown in Table 5. For LSTM on the left part, Alternating achieves excellent performance. By only 2-bit quantization on weights and activations, it exceeds Refined with 3-bit. The 2-bit result is even better than that reported in (Xie et al., 2017), where LSTM adding noising schemes for regularization can only attain 110.6 testing PPW. For GRU on the right part, although Alternating is much better than Refined, the 3-bit quantization still has gap with full precision one. We attribute that to the unified setting of hyper-parameters across all experiments. With specifically tuned hyper-parameters on this dataset, one may make up for that gap.

Note that our alternating quantization is a general technique. It is not only suitable for language models here. For a comprehensive verification, we apply it to image classification tasks. In both RNNs and feedforward neural networks, our alternating quantization also achieves the lowest testing error among all compared methods. Due to space limitation, we deter the results to Appendix B.

## 6    Conclusions

In this work, we address the limitations of RNNs, i.e., large memory and high latency, by quantization. We formulate the quantization by minimizing the approximation error. Under the key observation that some parameters can be singled out when others fixed, a simple yet effective alternating method is proposed. We apply it to quantize LSTM and GRU on language models. By 2-bit weights and activations, we achieve only a reasonably accuracy loss compared with full precision one, with $\sim 16\times$ reduction in memory and $\sim 6\times$ real acceleration on CPUs. By 3-bit quantization, we can attain compatible or even better result than the full precision one, with $\sim 10.5\times$ reduction in memory and $\sim 3\times$ real acceleration. Both beat existing works with a large margin. We also apply our alternating quantization to image classification tasks. In both RNNs and feedforward neural networks, the method can still achieve very plausible performance.

## 7    Acknowledgements

We would like to thank the reviewers for their suggestions on the manuscript. Zhouchen Lin is supported by National Basic Research Program of China (973 Program) (grant no. 2015CB352502), National Natural Science Foundation (NSF) of China (grant nos. 61625301 and 61731018), Qualcomm, and Microsoft Research Asia. Hongbin Zha is supported by Natural Science Foundation (NSF) of China (No. 61632003).

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

# APPENDIX

## A  BINARY MATRIX VECTOR MULTIPLICATION IN CPUS

Table 6: Computing time of the binary matrix vector multiplication in CPUs, where Quant represents the cost to execute our alternating quantization on-line.

| Weight Size | W-Bits / A-Bits | Total (ms) | Quant (ms) | Quant / Total | Acceleration |
|---|---|---|---|---|---|
| | 2/2 | 0.35 | 0.07 | 20% | 5.6× |
| $4096 \times 1024$ | 3/3 | 0.72 | 0.11 | 15% | 2.7× |
| | FP/FP | 1.95 | − | − | 1.0× |
| | 2/2 | 3.17 | 0.07 | 2.2% | 6.0× |
| $42000 \times 1024$ | 3/3 | 6.46 | 0.11 | 1.7% | 3.0× |
| | FP/FP | 19.10 | − | − | 1.0× |

In this section, we discuss the implementation of the binary multiplication kernel in CPUs. The binary multiplication is divided into two steps: Entry-wise XNOR operation (corresponding to entry-wise product in the full precision multiplication) and bit count operation for accumulation (corresponding to compute the sum of all multiplied entries in the full precision multiplication). We test it on Intel Xeon E5-2682 v4 @ 2.50 GHz CPU. For the XNOR operation, we use the Single instruction, multiple data (SIMD) $\_mm256\_xor\_ps$, which can execute 256 bit simultaneously. For the bit count operation, we use the function $\_popcnt64$ (Note that this step can further be accelerated by the up-coming instruction $\_mm512\_popcnt\_epi64$, which can execute 512 bits simultaneously. Similarly, the XNOR operation can also be further accelerated by the up-coming $\_mm512\_xor\_ps$ instruction to execute 512 bits simultaneously). We compare with the much optimized Intel Math Kernel Library (MKL) on full precision matrix vector multiplication and execute all codes in the single-thread mode. We conduct two scales of experiments: a matrix of size $4096 \times 1024$ multiplying a vector of size 1024 and a matrix of size $42000 \times 1024$ multiplying a vector of size 1024, which respectively correspond to the hidden state product $\mathbf{W}_h \mathbf{h}_{t-1}$ and the softmax layer $\mathbf{W}_s \mathbf{h}_t$ for Text8 dataset during inference with batch size of 1 (See Eq. (6)). The results are shown in Table 6. We can see that our alternating quantization step only accounts for a small portion of the total executing time, especially for the larger scale matrix vector multiplication. Compared with the full precision one, the binary multiplication can roughly achieve 6× acceleration with 2-bit quantization and 3× acceleration with 3-bit quantization. Note that this is only a simple test on CPU. Our alternating quantization method can also be extended to GPU, ASIC, and FPGA.

## B  IMAGE CLASSIFICATION

**Sequential MNIST.** As a simple illustration to show that our alternating quantization is not limited for texts, we conduct experiments on the sequential MNIST classification task (Cooijmans et al., 2017). The dataset consists of a training set of 60K and a test set of 10K $28 \times 28$ gray-scale images. Here we divide the last 5000 training images for validation. In every time, we sequentially use one row of the image as the input ($28 \times 1$), which results in a total of 28 time steps. We use 1 hidden layer's LSTM of size 128 and the same optimization hyper-parameters as the language models. Besides the weights and activations, the inputs are quantized. The testing error rates for 1-bit input, 2-bit weight, and 2-bit activation are shown in 7, where our alternating quantized method still achieves plausible performance in this task.

**MLP on MNIST.** The alternating quantization proposed in this work is a general technique. It is not only suitable for RNNs, but also for feed-forward neural networks. As an example, we firstly conduct a classification task on MNIST and compare with existing work (Li et al., 2017). The method proposed in (Li et al., 2017) is intrinsically a greedy multi-bit quantization method. For fair comparison, we follow the same setting. We use the MLP consisting of 3 hidden layers of 4096 units and an L2-SVM output layer. No convolution, preprocessing, data augmentation or pre-training is

Table 7: Testing error rate of LSTM on MNIST with 1-bit input, 2-bit weight, and 2-bit activation.

| Methods | Testing Error Rate |
|---|---|
| Full Precision | 1.10 % |
| Refined (Guo et al., 2017) | 1.39 % |
| Alternating (ours) | **1.19** % |

Table 8: Testing error rate of MLP on MNIST with 2-bit input, 2-bit weight, and 1-bit activation.

| Methods | Testing Error Rate |
|---|---|
| Full Precision | 0.97 % |
| Greedy (reported in (Li et al., 2017)) | 1.25 % |
| Refined (Guo et al., 2017) | 1.22 % |
| Alternating (ours) | **1.13** % |

Table 9: Testing error rate of CNN on CIFAR-10 with 2-bit weight and 1-bit activation.

| Methods | Testing Error Rate |
|---|---|
| Full Precision (reported in (Hou et al., 2017)) | 11.90 % |
| XNOR-Net (1-bit weight & activation, reported in (Hou et al., 2017)) | 12.62 % |
| Refined (Guo et al., 2017) | 12.08 % |
| Alternating (ours) | **11.70** % |

used. We also use ADAM (Kingma & Ba, 2015) with an exponentially decaying learning rate and Batch Normalization (Ioffe & Szegedy, 2015) with a batch size 100. The testing error rates for 2-bit input, 2-bit weight, and 1-bit activation are shown in Table 8. Among all the compared multi-bit quantization methods, our alternating one achieves the lowest testing error.

**CNN on CIFAR-10.** We then conduct experiments on CIFAR-10 and follow the same setting as (Hou et al., 2017). That is, we use 45000 images for training, another 5000 for validation, and the remaining 10000 for testing. The images are preprocessed with global contrast normalization and ZCA whitening. We also use the VGG-like architecture (Simonyan & Zisserman, 2015):

$$(2 \times 128 \, \text{C3}) - \text{MP2} - (2 \times 256 \, \text{C3}) - \text{MP2} - (2 \times 512 \, \text{C3}) - \text{MP2} - (2 \times 1024 \, \text{FC}) - 10 \, \text{SVM}$$

where C3 is a $3 \times 3$ convolution layer, and MP2 is a $2 \times 2$ max-pooling layer. Batch Normalization, with a mini-batch size of 50, and ADAM are used. The maximum number of epochs is 200. The learning rate starts at 0.02 and decays by a factor of 0.5 after every 30 epochs. The testing error rates for 2-bit weight and 1-bit activation are shown in Table 9, where our alternating method again achieves the lowest test error rate among all compared quantization methods.

