# OpenReview forum: "Alternating Multi-bit Quantization for Recurrent Neural Networks"
_ICLR.cc/2018/Conference — Accept (Poster)_

### Official Review · AnonReviewer3 · 2017-11-27
**Nicely written paper**

**Rating:** 7
**Confidence:** 2

**Review:**


Summary of the paper
-------------------------------

The authors propose a new way to perform multi-bit quantization based on greedy approximation and binary search tree for RNNs. They first show how this method, applied to the parameters only, performs on pre-trained networks and show great performances compared to other existing techniques on PTB. Then they present results with the method applied to both parameters and activations during training on 3 NLP datasets, showing again great performances compared to existing technique.

Clarity, Significance and Correctness
--------------------------------------------------

Clarity: The paper is clearly written.

Significance: I'm not familiar with the quantization literature, so I'll let more knowledgeable reviewers evaluate this point.

Correctness: The paper is technically correct.

Questions
--------------

1. It would be nice to have those memory and speed gains for training as well. Is it possible to use those quantization methods to train networks from scratch, i.e. without using a pre-train model?

Pros
------

1. The paper defines clear goals and contributions.
2. Existing methods (and their differences) are clearly and concisely presented.
3. The proposed method is well explained.
4. The experimental setup shows clear results compared to the non-quantized baselines and other quantization techniques.

Cons
-------

1. It would be nice to have another experiment not based on text (speech recognition / synthesis, audio, biological signals, ...) to see how it generalizes to other kind of data (although I can't see why it wouldn't).

Typos
--------

1. abstract: "gate recurrent unit" -> "gated recurrent unit"
2. equation (6): remove parenthesis in c_(t-1)
3. section 4, paragraph 1: "For the weight matrices, instead of on the whole, we quantize them row by row." -> "We don't apply quantization on the full matrices but rather row by row."
4. section 4, paragraph 2: Which W matrix is it? W_h? (2x)

Note
-------

Since I'm not familiar with the quantization literature, I'm flexible with my evaluation based on what other reviewers with more expertise have to say.

---

> ### Author Response · Authors · 2017-12-25
> **Issues during training**
>
> The memory costs during training can mainly be divided into two parts: the weights and the activations for backpropagation. For the weights, as a full precision should be maintained (See Eq. (7)), they cannot be reduced. For the activations, as it is enough to maintain a quantized version for backpropagation, we can have memory gains in this part. The time costs during training can also be divided into two parts: the forward and backward pass. During the forward pass, as the most costly full precision multiplications are transformed into the much faster binarized multiplications, we can have speed-ups in this part. During the backward, as we need to compute a full precision gradient, no speed-ups can be achieved.
>
> We conduct experiments of training from scratch in the PTB dataset and observe that it would result in 1~2 PPW worse than using a pre-trained model. But when combining with the continuation technique, that is, setting the initial number of bit to be large, then gradually decreasing it during training, it will result in almost no loss or even slightly better on accuracy. In fact, using a pre-trained model can also be regarded as such continuation technique, but coarser and simpler.
>
> In section 4, paragraph 2, W do means W_h.
>
> We will address other small typos in the revised version. We are also conducting experiments on non-text data and will report the results if time permits.

---

> > ### Author Response · Authors · 2018-01-05
> > **Experiment on Sequential MNIST**
> >
> > We add an experiment on sequential MNIST classification task, see the comment "Experiments on CIFAR10 and Sequential MNIST".

---

> > > ### Comment · AnonReviewer3 · 2018-01-11
> > > **Thanks for the additional experiments!**
> > >
> > > Thanks for the additional experiments!

---

### Official Review · AnonReviewer2 · 2017-11-28
**multi-bit quantization method for recurrent neural networks**

**Rating:** 8
**Confidence:** 4

**Review:**

I have read the comments and clarifications from the authors. They have added extra experiments, and clarified the speed-ups concern raised by others. I keep my original rating of the paper.

---------------
ORIGINAL REVIEW:

This paper introduces a multi-bit quantization method for recurrent neural networks, which is built on alternating the minimization formulated by Guo et al. 2017 by first fixing the \alpha values and then finding the optimal binary codes b_i with a BST, to then estimate \alpha with the refined approximation by Guo et al. 2017, iteratively. The observation that the optimal binary code can be computed with a BST is simple and elegant.

The paper is easy to follow and the topic of reducing memory and speeding up computations for RNN and DNN is interesting and relevant to the community.

The overall contribution on model quantization is based on existing methods, which makes the novelty of the paper suffer a bit. Said that, applying it to RNN is a convincing and a strong motivation. Also, in the paper it is shown how the matrix multiplications of the quantized model can be speeded up using 64 bits operation in CPU. This is, not only saves memory storage and usage, but also on runtime calculation using CPU, which is an important characteristic when there are limited computational resources.

Results on language models show that the models with quantized weights with 3 bits obtain the same or even slightly better performance on the tested datasets with impressive speed-ups and memory savings.

For completeness, it would be interesting, and I would strongly encourage to add a discussion or even an experiment using feedforward DNN with a simple dataset as MNIST, as most of previous work discussed in the paper report experiments on DNN that are feedforward. Would the speed-ups and memory savings obtained for RNN hold also for feedforward networks?

---

> ### Author Response · Authors · 2017-12-25
> **Speed-ups and memory savings for RNNs and feedforward networks**
>
> As we are quantizing the weight and activation to reduce the most costly matrix multiplication to binary operation, it is of no difference for RNNs and feedforward networks when concerning the speed-ups and memory savings.
>
> Please refer to the replies to common issues for the experiments on MNIST.

---

### Official Review · AnonReviewer1 · 2017-12-05
**Nice approach; skeptical about speed claims**

**Rating:** 7
**Confidence:** 4

**Review:**

Revision:

The authors have addressed my concerns around the achievable speedup. I am increasing my score to 7.

Original Review:

The paper proposes a technique for quantizing neural network weight matrices by representing columns of weight matrices as linear combinations of binary (+1/-1) vectors. Given a weight vector, the paper proposes an alternating optimization procedure to estimate the set of k binary vectors and coefficients that best represent (in terms of MSE) the original vector. This yields a k-bit quantization. First, the coefficients/binary weights are initialized using a greedy procedure proposed in prior work. Then, the binary weights are updated using a clever binary search procedure, followed by updates to the coefficients. Experiments are conducted in an RNN context for some language modeling tasks.

The paper is relatively easy to read, and the technique is clearly explained. The technique is as far as I can tell novel, and does seem to represent an improvement over existing approaches for similar multi-bit quantization strategies.

I have a few questions/concerns. First, I am quite skeptical of many of the speedup calculations: These are rather delicate to do properly, and depend on the specific instructions available, SIMD widths, the number of ALUs present in a core, etc. All of these can easily shift numbers around by a factor of 2-8x. Without an implementation in hand, comparing against a well-optimized reference GEMM for full floating point, it's not clear how much faster this approach really would be in practice. Also, the online quantization of activations doesn't seem to be factored into the speedup calculations, and no benchmarks are provided demonstrating how fast the quantization is (unless I'm missing something). This is concerning since the claimed speedups aren't possible without the online quantization of actiations.

It would have been nice to have more discussion of/comparison with other approaches capable of 2-4 bit quantization, such as some of the recent work on ternary quantization, product quantization approaches, or at least scalar (per-dimension) k-means (non-uniform quantization).

Finally, the experiments are reasonable, but the choice of RNN setting isn't clear to me. It would have been easier to compare to prior work if the experiments also included some standard image classification tasks (e.g., CIFAR10).

Overall though, I think the paper does just enough to warrant acceptance.

---

> ### Author Response · Authors · 2017-12-25
> **Discussions on ternary quantization and vector quantization**
>
> Please refer to the replies to common issues on speedup.
>
> Ternary quantization [1] is an extension to the binary quantization with one more feasible state, 0. It does quantization by tackling $\min_{\alpha, t}\|w – \alpha* t \|_2^2$ with $t$ restricted to {-1,0,+1}. However, currently there is no efficient algorithm to solve this problem. Instead, Li et al. [1] suggested to empirically set the entries $w_i$ with absolute scales less than $0.7/n \|w\|_1$ to 0 (n is the number of entries) and binarize the left entries with a closed-form solution as discussed in our paper. In fact, ternary quantization is a special case of the 2-bit quantization in our paper, i.e., $\min_{\alpha_1,\alpha_2, b_1,b_2}\|w – \alpha_1* b_1 - \alpha_2 * b_2 \|_2^2$ with an additional constraint that $\alpha_1 = \alpha_2$. Thus our alternating multi-bit quantization method can easily extend to solve it.
>
> In parallel to our binarized quantization, vector quantization is applied to compress the weights for feedforward neural networks [2][3]. Different from ours where all weights are directly constraint to {-1, +1}, vector quantization learns a small codebook by applying k-means clustering to the weights or conducting product quantization. The weights are then reconstructed by indexing the codebook. It has been shown that by such a technique, the number of parameters can be reduced by an order of magnitude with limited accuracy loss [2]. It is possible that our mutli-bit quantized binary weight can be further compressed by using the product quantization. However, this is out-of-the scope of this paper and we leave it for future work.
>
> We will incorporate the above discussions in the revised version. As for the experiment on image classification tasks, we have done on MNIST (see the replies to common issues). We will also report the results on CIFAR10 if time permits.
>
> [1] Li, Fengfu et al. Ternary weight networks, arXiv:1605.04711.
> [2] Gong, Yunchao et al. Compressing Deep Convolutional Networks using Vector Quantization, arXiv:1412.6115
> [3] Han, Song et al. Deep compression: Compressing deep neural networks with pruning, trained quantization and huffman coding, ICLR 15.

---

> > ### Author Response · Authors · 2018-01-05
> > **Experiment on CIFAR10**
> >
> > We add an experiment on CIFAR10, see the comment "Experiments on CIFAR10 and Sequential MNIST".

---

### Public Comment · (anonymous) · 2017-12-04
**Projected CPU inference acceleration is incorrect.**

> In the current generation of CPUs, we can perform 64 binary operations in one clock of CPU...

> With 2-bit weights and activations, we achieve only a reasonably accuracy loss compared with full precision one, with ∼16× reduction in memory and potential ∼13.5× acceleration on CPUs.

This is incorrect. To do an full binary inner-product MAC (i.e. xnor, popcount, accumulate) on an Intel CPU with AVX2 (until AVX512's popcount is available), the peak is approximately 256 GOP/cycle or so via xnor (issues on 3 ports, 1/3 of a cycle) and a fast popcnt method (e.g. Harley-Seal for larger reductions, lookup for smaller reductions in https://github.com/WojciechMula/sse-popcount/blob/master/results/skylake/skylake-i7-6700-gcc5.3.0-avx2.rst, which work out at a little bit over 2 cycles/vector in the best case). An equivalent Intel CPU can execute 2 fp32 AVX2 FMAs per cycle, which is a throughput of 32 FLOPs/cycle. Ignoring AVX2 thermal throttling, 1bit/1bit inner products vs fp32/fp32 inner products are then sped up by a factor of about 10x or so (and lower for Skylake's AVX-512 FMAs).

Thus, when you have to do 4 binary inner products (for 2b/2b inner products), your ideal theoretical speedup is 10x / 4 = 2.5x or so. This is a much less attractive number than the 13.5x quoted.

This is a key issue with several of these binary/ultra-low-precision convolution papers (arguably stemming from XNOR-Net), in that they overestimate the performance of binary operations on CPUs (either Intel or ARM), and underestimate int8/fp32 arithmetic throughput.

---

> ### Author Response · Authors · 2017-12-25
> **Issues on speed-ups**
>
> Please refer to the reply to common issues.

---

### Public Comment · ~Stephen_Merity1 · 2017-12-08
**Minor correction: Wikidata-2 => WikiText-2**

Interesting work :) Just wanted to briefly note that the official name for the Wikipedia language modeling dataset is WikiText-2 rather than Wikidata.

(I'd have submitted this just to the authors but there is no option for that within the comment posting)

---

> ### Author Response · Authors · 2017-12-25
> **Thank you**
>
> We will correct it in the revised version.

---

### Author Response · Authors · 2017-12-25
**Clarifications for issues concerned in common**

We thank all the reviewers for being positive towards our paper. Below are some clarifications for issues concerned in common:

Q1: “Acceleration for binary multiplication on CPU, both in theory and real implementation”

Reply: As the binary multiplication operates in 1 bit, whereas the full precision multiplication operates in 32 bit, despite the feasible implementations, the acceleration should be 32x in theory (Not 64x as claimed in XNOR-NET[1], the acceleration claimed in our paper was calculated based on this wrong factor. We will correct it in the revised version). In addition to binary operation, in real implementations, the acceleration can be largely affected by the size of the matrix, where much memory reduce can result in better utilizing in the limited cache (it is much faster than CPU main memory).

In this work, we implement the binary multiplication kernel in CPUs ourselves. The binary multiplication is divided into two steps: Entry-wise XNOR operation (corresponding to entry-wise product in the full precision multiplication) and bit count operation for accumulation (corresponding to compute the sum of all multiplied entries in the full precision multiplication). We test it on Intel Xeon E5-2682 v4 @ 2.50GHz CPU. For the XNOR operation, we use the Single instruction, multiple data (SIMD) _mm256_xor_ps, which can execute 256 bit simultaneously. For the bit count operation, we use the function _popcnt64 (Note that this step can further be accelerated by the up-coming instruction _mm512_popcnt_epi64, which can execute 512 bits simultaneously. Similarly, the XNOR operation can also be further accelerated by the up-coming _mm512_xor_ps instruction to execute 512 bits simultaneously). We compare with the much optimized Intel Math Kernel Library (MKL) on full precision matrix vector multiplication and execute all codes in the single-thread mode. We conduct two scales of experiments: a matrix of size $4096 \times 1024$ multiplying a vector of size $1024 \times 1$ and a matrix of size $42000 \times 1024$ multiplying a vector of size $1024 \times 1$, which respectively correspond to the hidden state product $W_h h_{t-1}$ and the softmax layer $W_s h_t$ for Text8 dataset during inference with batch size =1 (See Eq. (6) in the paper).

For a matrix of size $4096 \times 1024$ multiplying a vector of size $1024 \times 1$, we have

Full precision:   1.95ms
2-bit:          0.35ms (including 0.07ms for on-line quantizing the vector, taking 20%)
3-bit:          0.72ms (including 0.11ms for on-line quantizing the vector, taking 15%)

in which our 2-bit quantization has 5.6x acceleration and our 3-bit quantization has 2.7x acceleration.

For a matrix of size $42000 \times 1024$ multiplying a vector of size $1024 \times 1$, we have

Full precision:  19.10ms
2-bit:          3.17ms (including 0.07ms for on-line quantizing the vector, taking 2%)
3-bit:          6.46ms (including 0.11ms for on-line quantizing the vector, taking 1.7%)

in which our 2-bit quantization has 6x acceleration and our 3-bit quantization has 3x acceleration.

Note that this is only a simple test on CPU. Our alternating quantization method can also be extended to GPU, ASIC, and FPGA.

Finally, we deem that it may be too demanding to compare the speedups by pushing the limit of implementation. The exact number of speedups may vary across different computing devices and also depends on how much the compliers can be optimized. Our research is anyway valuable by showing the theoretical potential and inspiring future exploration.

Q2: “Illustration of experiments using feedforward neural networks”:

Reply: We conduct a classification task on MNIST and compare with existing work [2]. Besides the weights and activations, the input images are also quantized. The method proposed in [2] is intrinsically a greedy multi-bit quantization method. For fair comparison, we follow the same setting. We use the MLP consisting of 3 hidden layers of 4,096 units and an L2-SVM output layer. No convolution, preprocessing, data augmentation or pre-training is used. We also use ADAM with an exponentially decaying learning rate and Batch Normalization with a batch size 100. The testing error rates for 2 bit input, 2 bit weight, and 1 bit activation are as follows:

Full Precision (our implementation):           0.97%
Alternating (our method):                            1.13%
Refined (our implementation):                    1.22%
Greedy (reported in [2]):                              1.25%

Among all the compared multi-bit quantization methods, our alternating one achieves the lowest the testing error.

We will add all the above discussions and experiments in the revised version.

[1] Rastegari, Mohammad, et al. Xnor-net: Imagenet classification using binary convolutional neural networks, ECCV 2016.
[2] Li, Zefan, et al. Performance Guaranteed Network Acceleration via High-Order Residual Quantization, ICCV 2017.

---

### Author Response · Authors · 2018-01-05
**Experiments on CIFAR10 and Sequential MNIST**

Q1:“Including some standard image classification tasks (e.g., CIFAR10)” by Reviewer1

Reply: We conduct experiments on CIFAR-10 and follow the same setting as [1]. That is, we use 45000 images for training, another 5000 for validation, and the remaining 10000 for testing. The images are preprocessed with global contrast normalization and ZCA whitening. We also use the VGG-like architecture:

(2×128C3)−MP2−(2×256C3)−MP2−(2×512C3)−MP2−(2×1024FC)−10SVM,

where C3 is a 3×3 convolution layer, and MP2 is a 2×2 max-pooling layer. Batch Normalization, with a mini-batch size of 50, and ADAM are used. The maximum number of epochs is 200. The learning rate starts at 0.02 and decays by a factor of 0.5 after every 30 epochs. The testing error rates for 2-bit weight and 1-bit activation are as follows:

Alternating (our method):                                                             11.70%
Refined (our implementation):                                                     12.08%
XNOR-Net (1-bit weight & 1-bit activation, reported in [1])    12.62%
Full Precision (reported in [1])                                                      11.90%
where our alternating quantization method achieves the lowest test error rate.

Q2: “Including another experiment not based on text (speech recognition / synthesis, audio, biological, signals, ...) to see how it generalizes to other kind of data” by Reviewer 3

Reply: As a simple illustration, we conduct experiments on the sequential MNIST (images of size 28×28) classification task [2]. In every time, we sequentially use one row of the image as the input (of size 28×1), which results in a total of 28 time steps. We use 1 hidden layer’s LSTM of size 128 and the same optimization hyper-parameters as the Language Models in our paper. The testing error rates for 1-bit input, 2-bit weight, and 2-bit activation are as follows:

Full Precision (our implementation)          1.10%
Alternating (our method)                            1.19%
Refined (our implementation)                    1.39%
where our alternating quantized method still achieves plausible performance in this task.

We will add all the above experiments in the revised version.

[1] Hou, Lu, et al. Loss-aware Binarization of Deep Networks, ICLR 2017.
[2] Cooijmans, Tim, et al. Recurrent Batch Normalization, ICLR 2017.

---

### Decision · Program_Chairs · 2018-01-29
**ICLR 2018 Conference Acceptance Decision**

**Decision:**

Accept (Poster)

**Comment:**

The reviewers unanimously agree that this paper is worth publication at ICLR. Please address the feedback of the reviewers and discuss exactly how the potential speed up rates are computed in the appendix. I speed up rates to be different for different devices.